ZMP-HH/25-11

# An integrable deformed Landau-Lifshitz model with particle production?

MARIUS DE LEEUW$^a$, ANDREA FONTANELLA$^a$ AND JUAN MIGUEL NIETO GARCÍA$^b$

$^a$ *School of Mathematics & Hamilton Mathematics Institute,*
*Trinity College Dublin, Ireland*

mdeleeuw[at]maths.tcd.ie

andrea.fontanella[at]tcd.ie

$^b$ *II. Institut für Theoretische Physik, Universität Hamburg,*
*Luruper Chaussee 149, 22761 Hamburg, Germany*

juan.miguel.nieto.garcia[at]desy.de

## Abstract

We discuss the continuum limit of a non-Hermitian deformation of the Heisenberg XXX spin chain. This model appeared in the classification of $4 \times 4$ solutions of the Yang–Baxter equation and it has the particular feature that the transfer matrix is non-diagonalisable. We show that the model is given by a Drinfeld twist of the XXX spin chain and its continuum limit is a non-unitary deformation of the Landau–Lifshitz model. We compute the tower of conserved charges for this deformed Landau–Lifshitz model and show that they are generated by a boost operator. We furthermore show that it gives a non-vanishing $1 \to 2$ S-matrix, where one of the outgoing particles has vanishing energy and momentum, and thus it does not fulfil the usual "no particle production" condition of integrability. We argue that this result is natural when looked from the point of view of the non-diagonalisability of the spin chain.

# Contents

---

# Introduction

Integrable theories, both classical and quantum, are of great importance, appearing in nearly every branch of physics. This is due to the fact that they can be exactly solved, giving us toy models for a better understanding of physical phenomena. Sadly, integrable models are hard to find, and identifying if a Lagrangian or a Hamiltonian is integrable is typically far from easy. In fact, it is usually easier to construct an integrable model ex nihilo by searching for solutions of the Yang-Baxter equation (either classical or quantum, see [2–7] for some examples), than checking if a given theory is integrable.

For classical theories, the Arnold-Liouville theorem gives us a straight and clear distinction between integrable and non-integrable theories, which, in practice, translates into the use of methods like Lax pairs or mastersymmetries to construct enough conserved charges. The situation for quantum theories is completely different. For field theories in 1+1 dimensions, the usual test for integrability is to check if the $n \to n$ particle S-matrix factorises into products of $2 \to 2$ S-matrices, and if the $n \to m$ S-matrices vanish for $n \neq m$ [8–10]. However, when using this criterion, we should keep in mind that the proof requires the theory to be Lorentz invariant and the excitations to be massive. In fact, there are many examples of quantum field theories with either Galilei invariance or

massless excitations that have particle production, see [11–13] and references therein for examples.

In this article, we study the continuum limit of the "Class 5" model, described in [1] in a search for all possible two-dimensional integrable spin chains. This model (together with the "Class 6" model described in the same article) is a deformation of the well studied Heisenberg XXX spin chain and has a non-diagonalisable Hamiltonian, so its continuum limit is a non-unitary deformation of the Landau–Lifshitz (LL) action [14]. We show that the deformation of the LL action associated to the Class 5 model has a non-vanishing $1 \rightarrow 2$ S-matrix, where one of the outgoing particles has vanishing energy and momentum, and thus it does not have to fulfil the "no particle production" condition to be integrable. We will argue that this result is natural when looked from the point of view of the non-diagonalisability of the spin chain.

The Class 5 and Class 6 are interesting integrable models because they represent two of the few examples of integrable spin chains that have a non-diagonalisable transfer matrix (and, thus, a non-diagonalisable Hamiltonian). Other examples of models with non-diagonalisable transfer matrix are the eclectic spin chain [15, 16], the open XXZ chain at roots of unity [17], and the q-deformed Haldane–Shastry model at $q = i$ [18, 19]. Although these theories are integrable, direct application of the usual Coordinate or Algebraic Bethe Ansatz cannot retrieve the full Hilbert space [16], forcing us to use alternative methods, like counting states with the Pólya enumeration theorem [20, 21]. Interestingly, in cases where the non-diagonalisable model is obtained as a limit of a diagonalisable integrable model, careful computation of the limit provides the full Hilbert space [22, 23] (a similar method was proposed for the open XXZ chain at roots of unity in [17], but it was not fully justified). Thus, there is hope that in the future the Bethe Ansatz will be improved to take care of this situation.

In addition to their importance by itself, the Class 5 model has a connection to the topic of Yang-Baxter deformations in AdS/CFT. The LL action played an important role as a non-trivial check of the duality between strings in $\mathrm{AdS}_5 \times \mathrm{S}^5$ and $\mathcal{N} = 4$ SYM. In particular, exactly the same generalization of the LL action appears both when computing the large angular momentum expansion around the solution of rotating strings in $\mathbb{R} \times \mathrm{S}^5$ with two large angular momenta and when computing the anomalous dimensions of operators of the form $\mathrm{Tr}(Z^{L-N} X^N)$ for large $L$ and $N$ [24,25] (which was later generalised to fermionic sectors [26,27], non-compact sectors [28], and sectors containing both [29–31]). We can pose a similar problem in the context of the Yang-Baxter deformation of $\mathrm{AdS}_5 \times \mathrm{S}^5$, where we use solutions of the classical Yang-Baxter equation to construct deformations of the $\mathrm{AdS}_5 \times \mathrm{S}^5$ background [32–36]. On the one hand, following the spirit of [24], we can study the deformed LL action obtained by expanding the string action around spinning string solutions, see for example, [37, 38] for the case of the beta deformation, [39, 40] for the eta deformation, [41] for the dipole deformation, and [42, 43] for the bi-Yang-Baxter

deformation. On the other hand, it was proposed that the deformation manifests in the dual field theory as a Drinfeld twist [44]. An example is the duality between string theory in the Lunin-Maldacena background and the beta deformation of $\mathcal{N} = 4$ SYM [45]. Recently, to gain a better insight into these deformations, [46] studied the case of a Jordanian deformation of the non-compact $\text{XXX}_{-\frac{1}{2}}$ spin chain [46]. As the Class 5 model can be written as a Jordanian Drinfeld twist of the XXX spin chain, it provides a simpler toy model where many of the unconventional characteristic of Jordanian deformations and non-diagonalisable spin chains can be studied (for example, particle production despite having integrability [47]).

Another situation where integrability and non-diagonalisability meet is in the context of non-relativistic strings. The equations of motion of strings propagating in the non-relativistic limit of $\text{AdS}_5 \times \text{S}^5$ can be written as zero-curvature equations of a Lax connection [48, 49]. However, the classical monodromy matrix associated to this Lax connection does not fit the standard form: it is non-diagonalisable and the eigenvalues are independent of the spectral parameter [50]. Although the Class 5 model is a quantum theory, it also has a non-diagonalisable transfer matrix, so it might shed some light into this problem.

The outline of this paper is as follows. In section 1 we briefly review the Class 5 model and its properties, showing that it can be written as a Drinfeld twist of the XXX spin chain. In section 2 we construct the continuum limit of the Class 5 Hamiltonian, giving us a deformed version of the LL action. We show that the continuum limit of the boost operator used in [1] to construct the tower of conserved charges of the model has a classical counterpart. This is enough to claim that the model is classically integrable. In section 3 we perform a field redefinition to transform the kinetic term into the standard form, making canonical quantization straightforward. Expanding the action in series in the number of fields, we show that the action has a non-vanishing $1 \to 2$ S-matrix and that the $2 \to 2$ S-matrix is not deformed. In section 4 we summarise our results and discuss some possible future directions. For completeness, we have included some similar computations for the Class 6 model in the appendix A.

# 1 The Class 5 model of [1]

The model is given by the simplest quantum integrable deformation of the XXX spin chain R-matrix, dubbed "Class 5" model, that appeared in the classification [1]. It is

given by[1]

$$
R_5(u) = \begin{pmatrix}
2a_1u + 1 & a_2u & -a_2u & a_2a_3u^2 \\
0 & 2a_1u & 1 & -a_3u \\
0 & 1 & 2a_1u & a_3u \\
0 & 0 & 0 & 2a_1u + 1
\end{pmatrix} ,
\tag{1.1}
$$

where $a_1, a_2, a_3$ are free parameters. The parameter $a_1$ is unphysical, as it can be eliminated by the redefinition: $u \to u/a_1$, $a_2 \to a_1a_2$, $a_3 \to a_1a_3$. From now on, we will set $a_1 = 1$.

The R-matrix (1.1) admits furthers simplifications, which will be used in this paper. We can use a local basis transformation to shift the parameter $a_3$. In particular, one can bring it to the form as if $a_2 = a_3$, by performing the local basis transformation

$$
R_5^{a_2=a_3} = [V(u) \otimes V(v)]R_5(u - v)[V(u) \otimes V(v)]^{-1}, \quad \text{where} \quad V(u) = \begin{pmatrix} 1 & -\frac{a_2-a_3}{2}u \\ 0 & 1 \end{pmatrix} .
\tag{1.2}
$$

Similarly, it can be brought to a form akin to having $a_3 = 0$ by using instead

$$
V(u) = \begin{pmatrix} 1 & a_3u \\ 0 & 1 \end{pmatrix} .
\tag{1.3}
$$

The R-matrix (1.1) satisfies the Yang-Baxter equation (YBE)

$$
R_{12}(u - v)R_{13}(u)R_{23}(v) = R_{23}(v)R_{13}(u)R_{12}(u - v) ,
\tag{1.4}
$$

and its corresponding quantum Hamiltonian is

$$
\mathcal{H}_5 = \sum_{j=1}^{L} h_{j,j+1} , \qquad h_{L,L+1} = h_{L,1} ,
\tag{1.5}
$$

$$
h_{j,j+1} = 2P_{j,j+1} + \frac{a_2 - a_3}{2} \left( \mathbb{I}_j \otimes \sigma_{j+1}^+ - \sigma_j^+ \otimes \mathbb{I}_{j+1} \right) + \frac{a_2 + a_3}{2} \left( \sigma_j^z \otimes \sigma_{j+1}^+ - \sigma_j^+ \otimes \sigma_{j+1}^z \right)
$$

$$
= \begin{pmatrix}
2 & a_2 & -a_2 & 0 \\
0 & 0 & 2 & a_3 \\
0 & 2 & 0 & -a_3 \\
0 & 0 & 0 & 2
\end{pmatrix}_{j,j+1} ,
$$

where $L$ is the length of the spin chain, and $P_{j,j+1}$ is the permutation operator of the sites $j$ and $j + 1$.

---

[1] The R-matrix given in (1.1) is the Class 5 R-matrix of [1] multiplied by $(1 - a_1u)^{-1}$, which amounts to shifting the Hamiltonian by a multiple of the identity operator, and is therefore unphysical.

One can immediately notice that the term with coefficient $\frac{1}{2}(a_2 - a_3)$ vanishes, due to the periodic boundary conditions of the spin chain. Then the Hamiltonian (1.5) describes just a one parameter deformation of the XXX spin chain. We will denote the deformation parameter by $a \equiv \frac{1}{2}(a_2 + a_3)$.

## 1.1   Realisation as a Drinfeld twist of the XXX spin chain

The R-matrix (1.1) can be written using Pauli matrices as

$$
R_5(u) = 2u\,\mathbb{I} \otimes \mathbb{I} + P_{1,2} + u\left(\frac{a_2 + a_3}{2}\mathbb{I} + \frac{a_2 - a_3}{2}\sigma^z\right) \otimes \sigma^+
$$
$$
- u\,\sigma^+ \otimes \left(\frac{a_2 + a_3}{2}\mathbb{I} + \frac{a_2 - a_3}{2}\sigma^z\right) + a_2 a_3 u^2 \sigma^+ \otimes \sigma^+ . \quad (1.6)
$$

A very common choice for an L-operator to apply the Algebraic Bethe Ansatz construction is $L = R$. In this case, it is easy to check that

$$
L_i(u) = \left(\frac{1}{2} + 2u\right)\mathbb{I} \otimes \mathbb{I} + \sum_\alpha \sigma^\alpha \otimes \mathfrak{s}^\alpha + u\left(\frac{a_2 + a_3}{2}\mathbb{I} + \frac{a_2 - a_3}{2}\sigma^z\right) \otimes \mathfrak{s}^+
$$
$$
- u\,\sigma^+ \otimes \left(\frac{a_2 + a_3}{2}\mathbb{I} + \frac{a_2 - a_3}{2}\mathfrak{s}^z\right) + a_2 a_3 u^2 \sigma^+ \otimes \mathfrak{s}^+ , \quad (1.7)
$$

only fulfils the RLL equation when the spin operators are the Pauli matrices, that is, $\mathfrak{s}^\alpha = \frac{1}{2}\sigma^\alpha$. Here $\mathfrak{s}^\alpha$ are $\mathfrak{su}(2)$ generators, which satisfy the algebra relation $[\mathfrak{s}^\alpha, \mathfrak{s}^\beta] = \varepsilon_{\alpha\beta\gamma}\mathfrak{s}^\gamma$. Notice that we are distinguishing operators acting on the auxiliary space and operators acting on the physical space by using the Pauli matrices $\sigma^\alpha$ for the former but keeping a generic representation $\mathfrak{s}^\alpha$ for the latter.

Despite that, there is a way to write an L-operator that works for any representation of the $\mathfrak{su}(2)$ algebra. To find it, we first should notice that the R-matrix (1.1) for $a_3 = 0$, which can be obtained by the basis rotation given in (1.3), can be decomposed as follows

$$
R_5^{a_3=0}(u) = \begin{pmatrix} 1 & \frac{a_2}{2} & 0 & 0 \\ 0 & 1 & 0 & 0 \\ 0 & 0 & 1 & 0 \\ 0 & 0 & 0 & 1 \end{pmatrix} \begin{pmatrix} 2u+1 & 0 & 0 & 0 \\ 0 & 2u & 1 & 0 \\ 0 & 1 & 2u & 0 \\ 0 & 0 & 0 & 2u+1 \end{pmatrix} \begin{pmatrix} 1 & 0 & -\frac{a_2}{2} & 0 \\ 0 & 1 & 0 & 0 \\ 0 & 0 & 1 & 0 \\ 0 & 0 & 0 & 1 \end{pmatrix} . \quad (1.8)
$$

We notice that the third matrix, written as

$$
F = \mathbb{I} \otimes \mathbb{I} + \frac{a_2}{2}\mathfrak{s}^+ \otimes \left(\frac{1}{2}\mathbb{I} + \mathfrak{s}^z\right) , \quad (1.9)
$$

fulfils the 2-cocycle condition with a trivial coproduct (that is, $\Delta(\mathfrak{s}^\alpha) = \mathfrak{s}^\alpha \otimes \mathbb{I} + \mathbb{I} \otimes \mathfrak{s}^\alpha$)

$$F_{12}(\Delta \otimes \mathbb{I})F = F_{23}(\mathbb{I} \otimes \Delta)F \ . \tag{1.10}$$

This means that the Class 5 R-matrix can be obtained as a Drinfeld twist of the XXX spin chain R-matrix, since the equation (1.8) can be written as

$$R_5^{a_3=0}(u) = F_{21} R_{XXX}(u) F_{12}^{-1} \ . \tag{1.11}$$

It is then straightforward to check that the L-operator obtained from this twisting

$$L_5(u) = \begin{pmatrix} (\mathbb{I} + a_2\mathfrak{s}^+)(2u + \mathfrak{s}^{11}) & \mathfrak{s}^- - a_2(2u + \mathfrak{s}^{11})\mathfrak{s}^{11} + a_2\mathfrak{s}^+(\mathfrak{s}^+ - a_2(2u + \mathfrak{s}^{11})\mathfrak{s}^{11}) \\ \mathfrak{s}^+ & 2u + \mathfrak{s}^{22} - a_2\mathfrak{s}^+\mathfrak{s}^{11} \end{pmatrix} ,$$
$$\tag{1.12}$$

where $\mathfrak{s}^{11} \equiv \frac{1}{2}\mathbb{I} + \mathfrak{s}^z$ and $\mathfrak{s}^{22} \equiv \frac{1}{2}\mathbb{I} - \mathfrak{s}^z$ to shorten the expressions, can be proven to fulfil the RLL equation with the R-matrix of the Class 5 model.

# 2    The continuum limit

We are now interested in taking the continuum limit of the model described in section 1, namely the limit in which the lattice spacing of the spin chains goes to zero. By doing that, we will transition from the *quantum* description of the spin chain, given by an R-matrix, to its *classical* description, given by a field theory. We will follow the procedure described by Fradkin in his book [51].

The idea is to introduce coherent states for $\mathfrak{su}(2)$. For a given spin $s$ representation, there are $2s + 1$ states, which are constructed by acting with the ladder operators $\mathfrak{s}_\pm$ on the highest weight state, which we denote by $|0\rangle \equiv |s, s\rangle$. The highest weight state is an eigenstate of both $\mathfrak{s}_3$ and the quadratic Casimir invariant $\vec{\mathfrak{s}}^2$ that satisfies the relations

$$\mathfrak{s}_3 |0\rangle = s |0\rangle \ , \tag{2.1}$$
$$\vec{\mathfrak{s}}^2 |0\rangle = s(s + 1) |0\rangle \ . \tag{2.2}$$

In this work we will take the spin $s = 1/2$ representation, i.e. $\mathfrak{s}^\alpha = \frac{1}{2}\sigma^\alpha$, and the highest weight state is given by $|0\rangle = (1, 0)$.

The coherent states are obtained via a rotation of the highest weight state as follows [52]

$$|\theta, \phi\rangle = e^{i\mathfrak{s}_3\phi} e^{i\mathfrak{s}_2\theta} |0\rangle \ . \tag{2.3}$$

These states are labelled by two angles, $\phi$ and $\theta$, which will gain the meaning of fields

of a 2d sigma model. The action that captures the continuum limit of the quantum Hamiltonian (1.5) is given by

$$S_5 = \int \mathrm{d}t \mathrm{d}x \left( \mathcal{L}_{\mathrm{kin}} + \langle \theta, \phi | \otimes \langle \theta + \delta\theta, \phi + \delta\phi | \, h_{12} \, |\theta, \phi\rangle \otimes |\theta + \delta\theta, \phi + \delta\phi\rangle \right), \quad (2.4)$$

where the displacements $\delta\theta$ and $\delta\phi$ are given by the Taylor expansion for an infinitesimal lattice spacing $\delta x \equiv \varepsilon$, as

$$\delta\theta = \varepsilon\theta_x + \frac{1}{2}\varepsilon^2\theta_{xx} + \mathcal{O}(\varepsilon^3), \qquad\qquad \delta\phi = \varepsilon\phi_x + \frac{1}{2}\varepsilon^2\phi_{xx} + \mathcal{O}(\varepsilon^3). \quad (2.5)$$

The kinetic term that appears by taking the $SU(2)$ coherent state continuum limit does not depend on the specific R-matrix. Its expression is universal, given by

$$\mathcal{L}_{\mathrm{kin}} = -\varepsilon^2 \cos\theta \, \phi_t. \quad (2.6)$$

Here we denoted time and space derivatives as $\partial_t X \equiv X_t$, $\partial_x X \equiv X_x$, a convention that we will use throughout the paper. By plugging the expression for $h_{12}$ given in (1.5), we get

$$S_5 \; = \; \int \mathrm{d}t \mathrm{d}x \left[ 2 + a\left(\varepsilon\mathcal{A}^{(1)} + \varepsilon^2\mathcal{A}^{(2)}\right) + b\left(\varepsilon\mathcal{B}^{(1)} + \varepsilon^2\mathcal{B}^{(2)}\right) + \varepsilon^2\mathcal{L}_{\mathrm{LL}} + \mathcal{O}(\varepsilon^3) \right], \quad (2.7)$$

where

$$\mathcal{A}^{(1)} \;=\; -\frac{e^{-i\phi}}{2}\left(\theta_x - \frac{i}{2}\sin 2\theta \, \phi_x\right), \quad (2.8)$$

$$\mathcal{A}^{(2)} \;=\; \frac{e^{-i\phi}}{2}\left(i\cos^2\theta \, \theta_x\phi_x + \frac{1}{4}\sin 2\theta \, (\phi_x)^2 - \frac{1}{2}\theta_{xx} + \frac{i}{4}\sin 2\theta \, \phi_{xx}\right), \quad (2.9)$$

$$\mathcal{B}^{(1)} \;=\; -\frac{e^{-i\phi}}{2}\left(\cos\theta \, \theta_x - i\sin\theta \, \phi_x\right), \quad (2.10)$$

$$\mathcal{B}^{(2)} \;=\; \frac{e^{-i\phi}}{4}\left(\sin\theta \, (\theta_x)^2 + 2i\cos\theta \, \theta_x\phi_x + \sin\theta \, (\phi_x)^2 - \cos\theta \, \theta_{xx} + i\sin\theta \, \phi_{xx}\right), (2.11)$$

$$\mathcal{L}_{\mathrm{LL}} \;=\; -\cos\theta \, \phi_t - \frac{1}{2}\left((\theta_x)^2 + \sin^2\theta (\phi_x)^2\right), \quad (2.12)$$

where $a \equiv \frac{1}{2}(a_2 + a_3)$, and $b \equiv \frac{1}{2}(a_2 - a_3)$. The deformation term proportional to $b$ is a total derivative, in agreement with the observation that this term cancels out in the quantum Hamiltonian due to periodic boundary conditions. Furthermore, $\mathcal{A}^{(2)}$ turns out to be a total derivative as well. Therefore, from now on, we disregard $\mathcal{A}^{(2)}, \mathcal{B}^{(1)}, \mathcal{B}^{(2)}$ and the constant term from the action (2.7). To understand the dynamics of this model, one possibility is to treat $\mathcal{A}^{(1)}$ and $\mathcal{L}_{\mathrm{LL}}$ as independent actions, and derive their equations of motion. This will produce the usual LL equations of motion, plus some extra equations

that can be understood as constraints. However, it turns out that this method gives a sort of trivial one-dimensional model.[2] Another issue of this procedure is that it is unclear at which order in $\varepsilon$ one should stop to generate equations of motion, or if all of them, which are infinite, should be considered.

A second possibility, which is the one that we will follow, is to bring (2.7) into the form of a homogeneous action of order $\varepsilon^2$. We do this by rescaling the deformation parameter as $a = \varepsilon\alpha$. In this way, we find that the sigma model associated to the Class 5 R-matrix of [1] is a LL model with a non-unitary deformation, given by the leading term of (2.7) in the $\varepsilon \to 0$ limit:

$$S_5^{\text{lead}} = -\varepsilon^2 \int \mathrm{d}t\mathrm{d}x \left[ \cos\theta\,\phi_t + \frac{1}{2}\left[(\theta_x)^2 + \sin^2\theta(\phi_x)^2\right] + \alpha\,\frac{e^{-i\phi}}{2}\left(\theta_x - \frac{i}{2}\sin 2\theta\,\phi_x\right)\right] \quad (2.13)$$

This action can equivalently be written in terms of the spin density $S_j(t,x)$, $j = 1, 2, 3$, under the constraint that the length of the spin is 1, i.e. $\sum_{j=1}^3 (S_j)^2 = 1$. Using the map,

$$\vec{S} = \begin{pmatrix} \sin(\theta)\cos(\phi) \\ \sin(\theta)\sin(\phi) \\ \cos(\theta) \end{pmatrix}, \quad (2.14)$$

the classical Hamiltonian associated to (2.13) is

$$H_5 = \int \mathrm{d}x \left[ \frac{1}{2}\vec{S}_x^T \vec{S}_x + \frac{\alpha}{2}\vec{S}^T M \vec{S}_x \right], \qquad M \equiv \begin{pmatrix} \beta & 0 & -1 \\ 0 & \beta & i \\ 1 & -i & \beta \end{pmatrix}, \quad (2.15)$$

where $\beta$ entering in the matrix $M$ is a free parameter. This parameter is spurious since $\vec{S}_x^T \vec{S} = 0$ and hence the Hamiltonian does not depend on $\beta$ and we will set it to 0 from now on.

An alternative way to write this action is by introducing the generalised derivative,

$$D_x \vec{S} = \vec{S}_x - \frac{\alpha}{2} M \vec{S}, \quad (2.16)$$

---

[2]In more detail, the equations of motion generated from $\mathcal{A}^{(1)}$ are:

$$e^{-i\phi}\sin^2\theta\,\phi_x = 0, \qquad e^{-i\phi}\sin^2\theta\,\theta_x = 0.$$

There are two possibility to solve these equations. The first one consists in restricting the fields to only dependent on $t$, i.e. $\phi = \phi(t)$ and $\theta = \theta(t)$. Then the LL equations from $\mathcal{L}_{\text{LL}}$ impose that $\theta_t = \phi_t = 0$, which is solved by the constant function, and therefore the dynamics is trivial. The second possibility consists in taking $\theta = k\pi$, with $k \in \mathbb{Z}$, while $\phi$ is unconstrained. The fact that $\phi$ is unconstrained is not a problem because $\theta = k\pi$ are the north and south poles of the sphere, where the angle $\phi$ plays no role. The dynamics is again trivial for this solution.

then the Hamiltonian takes the form

$$H_5 = \int dx \left[ \frac{1}{2}(D_x \vec{S})^T D_x \vec{S} - \frac{\alpha^2}{8} \vec{S}^T M^T M \vec{S} \right] . \tag{2.17}$$

This looks like a Hamiltonian of a massive field in the presence of some sort of gauge field. However, the mass-like term is complex since

$$M^T M = \begin{pmatrix} 1 & -i & 0 \\ -i & -1 & 0 \\ 0 & 0 & 0 \end{pmatrix} = \begin{pmatrix} 0 & i & i \\ 0 & 1 & 0 \\ 1 & 0 & 0 \end{pmatrix} \begin{pmatrix} 0 & 0 & 0 \\ 0 & 0 & 1 \\ 0 & 0 & 0 \end{pmatrix} \begin{pmatrix} 0 & 0 & 1 \\ 0 & 1 & 0 \\ -i & -1 & 0 \end{pmatrix} . \tag{2.18}$$

It is interesting to note that in this formalism the mass matrix is non-diagonalisable and has Jordan normal form with 0s everywhere and 1 on one entry in the upper diagonal.

We can transform this generalised derivative into a regular derivative by performing the field redefinition[3]

$$\vec{S} = \exp\left(\frac{\alpha}{2}xM\right)\vec{\mathbb{S}}, \tag{2.19}$$

and in terms of the new variable $\vec{\mathbb{S}}$ the Hamiltonian reads

$$H_5 = \int dx \left[ \frac{1}{2}\vec{\mathbb{S}}_x^T \vec{\mathbb{S}}_x - \frac{\alpha^2}{8}\vec{\mathbb{S}}^T M^T M \vec{\mathbb{S}} \right] . \tag{2.20}$$

As the mass matrix $M^T M$ is not diagonalisable, this model cannot be related by a rotation to the anisotropic LL theory.

The dynamics of the system is fixed by the classical Hamiltonian and by the Poisson brackets

$$\{S^i(t,x), S^j(t,y)\} = \varepsilon_{ijk}S^k(t,x)\delta(x-y), \tag{2.21}$$

which produce the following equations of motion

$$\vec{S}_t = \vec{S} \times \left( \vec{S}_{xx} - \alpha M \vec{S}_x \right) . \tag{2.22}$$

Although this Hamiltonian is obtained from the continuum limit of an integrable model, this is not enough to guarantee the integrability of the classical model. Nevertheless, we can construct a (component of the) Lax matrix with the appropriate Poisson structure. Applying the previous procedure to the quantum Lax (1.12) is immediate because coherent states are constructed in such a way that spin operators $\mathfrak{s}^i$ acting on them

---

[3]We thank B. Hoare for this observation.

gives us the classical spin densities $S^i$

$$\langle -\theta, -\phi | \, \mathfrak{s}^i \, | -\theta, -\phi \rangle = S^i \,, \qquad (2.23)$$

Thus, a classical Lax can be obtained from (1.12) by just replacing $\mathfrak{s}^i$ by $S^i$. Using the Poisson brackets (2.21), it can be checked that the Lax matrix obtained by this procedure has a Kostant-Kirillov Poisson bracket

$$\{L_1(u), L_2(v)\} = [r_{12}(u-v), L_1(u)L_2(v)] \,, \qquad (2.24)$$

where the classical R-matrix is given by the appropriate limit of the classical one

$$r_{12}(u) = -\left( \frac{R_5(u)}{2u} - \mathbb{I} \otimes \mathbb{I} \right) \,. \qquad (2.25)$$

Sadly, we were not able to find a companion matrix to check that we can extract the equations of motion from this Lax matrix, and thus show that the action is integrable. Nevertheless, in the following sections we will show that the action (2.13) admits an infinite tower of conserved charges in involution. In particular, we will also show that they can be constructed though a strong symmetry.

## 2.1 Spacetime symmetries

The deformation term entering in (2.13) introduces terms of order one in the spatial derivative of the fields. In principle, this may affect the spacetime symmetries of the action, which we are going to analyse in this section.

It is convenient to first identify the spacetime symmetries of the LL action, i.e. when we first set $\alpha = 0$. We consider the following transformation of the spacetime coordinates,

$$\delta t = \xi^t \,, \qquad\qquad \delta x = \xi^x \,, \qquad (2.26)$$

and we write down the most general linear transformation of the fields:

$$\delta\theta = \xi^t \theta_t + \xi^x \theta_x + \gamma_1 \, \theta + \gamma_2 \, \phi \,, \qquad (2.27\text{a})$$
$$\delta\phi = \xi^t \phi_t + \xi^x \phi_x + \gamma_3 \, \phi + \gamma_4 \, \theta \,, \qquad (2.27\text{b})$$

where $\gamma_i$ are generic functions of the spacetime coordinates. We demand that the LL Lagrangian transforms as a total derivative under the field transformation (2.27). This

imposes the following conditions,

$$\gamma_i = 0\,, \qquad \forall\, i = 1, ..., 4\,, \tag{2.28a}$$

$$\xi^t = c_t\,, \qquad \xi^x = c_x\,, \tag{2.28b}$$

which means that the only spacetime symmetries are the time and space translations, generated by $H = \partial_t$ and $P = \partial_x$ respectively. In particular, the theory does not admit Galilean boosts, therefore it is Aristotelian.

In addition to these symmetries of the action, called "off-shell" symmetries, there is also another important symmetry realised only at the level of the equations of motion, namely "on-shell", given by the *anisotropic dilatation*

$$t \to \lambda^z t\,, \qquad x \to \lambda x\,. \tag{2.29}$$

For the LL theory, the scaling coefficient $z$ needs to be fixed as $z = 2$. This is a symmetry only on-shell, since the action transforms under it as $S_{LL} \to \lambda S_{LL}$.

Now we turn on the deformation, i.e. $\alpha \neq 0$, and we ask whether the symmetries of the LL model are preserved by the full deformed action. It turns out that the off-shell symmetries generated by $H$ and $P$ are preserved, but the on-shell anisotropic dilatation symmetry is broken due to the fact the deformation term is homogeneous of order one in the spatial derivatives.

## 2.2 Higher conserved charges

Integrable models that are described in terms of an R-matrix admit an elegant procedure to construct the infinite tower of conserved charges. This procedure makes use of the so-called *boost operator*[4] $\mathcal{B}[\mathcal{H}]$ [53], see e.g. [54] for a review. For a spin-chain of length $L$ with nearest neighbour interactions described by the R-matrix $R(u)$, one can define the Hamiltonian $\mathcal{H}$, or $\mathcal{Q}_2$, as

$$\mathcal{Q}_2 \equiv \sum_{n=1}^{L} h_{n,n+1} = \sum_{n=1}^{L} R_{n,n+1}^{-1} \frac{\mathrm{d}}{\mathrm{d}u} R_{n,n+1} \bigg|_{u=0}\,, \tag{2.30}$$

which is a conserved charge of range 2. Then the boost operator is defined as

$$\mathcal{B}[\mathcal{H}] \equiv \sum_{n=-\infty}^{+\infty} n\, h_{n,n+1}\,, \tag{2.31}$$

---

[4]Not to be confused with the Galiean boost, which is not a symmetry of this model.

and the conserved charges of range $r + 1$ is obtained recursively by commuting the boost operator with the conserved charge of range $r$,[5]

$$\mathcal{Q}_{r+1} = [\mathcal{B}[\mathcal{H}], \mathcal{Q}_r], \qquad (2.32)$$

with the property that all these higher conserved charges commute among each others, i.e. $[\mathcal{Q}_p, \mathcal{Q}_q] = 0$, for all $p$ and $q$.

Now, one may wonder if this elegant construction of higher conserved charges has any analogue in the continuum limit. The answer is yes, and for the LL theory this has already been studied in [55]. The idea is that in the continuum limit the system is described, e.g. by using the coherent state method, by a classical Hamiltonian $H$, or $Q_2$,

$$H = \int_0^L \mathrm{d}x \, \mathcal{H}, \qquad (2.33)$$

$$\mathcal{H} = \left[ (\theta_x)^2 + \sin^2 \theta (\phi_x)^2 \right] + \alpha \frac{e^{-i\phi}}{2} \left( \theta_x - \frac{i}{2} \sin 2\theta \, \phi_x \right), \qquad (2.34)$$

and the analogue of the boost operator is given by the functional $B[H]$, defined as

$$B[H] \equiv \int_{-\infty}^{+\infty} \mathrm{d}x \, x \, \mathcal{H}. \qquad (2.35)$$

We can check, for example, that the quantity

$$Q_3 = \{B[H], H\} = \int_0^L \mathrm{d}x \, \mathcal{Q}_3, \qquad (2.36)$$

$$\mathcal{Q}_3 = \frac{1}{2} \left[ 4 \cos \theta \, (\theta_x^2 \phi_x) + \sin \theta \left( \sin(2\theta) \, \phi_x^2 - 2 \, \theta_{xx} \right) \phi_x + 2 \sin \theta \, (\theta_x \phi_{xx}) \right]$$

$$+ \frac{i}{8} \alpha e^{-i\phi} \left[ 4 \cos \theta \, (\theta_{xx} - 2i \, \theta_x \phi_x) + 2 \sin \theta \left( 4 \, \theta_x^2 + [1 - 3 \cos(2\theta)] \, \phi_x^2 - 2i \, \phi_{xx} \right) \right]$$

$$+ \frac{\alpha^2}{2} e^{-2i\phi} \sin \theta \, (i\theta_x + \cos \theta \sin \theta \, \phi_x). \qquad (2.37)$$

is actually a conserved charge of our action (2.13). In the spin variables, $\mathcal{Q}_3$ reads

$$\mathcal{Q}_3 = \varepsilon_{ijk} \left[ S^i S_x^j S_{xx}^k + \frac{\alpha}{2} \left( 2 S^i M_{jl} S_x^l S_x^k - S^i M_{jl} S^l S_{xx}^k \right) + \frac{\alpha^2}{2} M_{il} S^l M_{jm} S_x^m S^k \right], \qquad (2.38)$$

---

[5]Technically speaking, the boost operator is defined on a spin-chain of infinite length, whereas the higher charges are defined on a spin-chain of length $L$. However, once can check that $[\mathcal{B}[\mathcal{H}], \mathcal{Q}_r]$ is an operator of order $r + 1$ and therefore well-defined on a finite spin chain.

and it can be rewritten in a more compact form in terms of the generalised derivative,

$$\mathscr{Q}_3 = \varepsilon_{ijk} \left[ S^i D_x S^j D_x^2 S^k + \frac{\alpha^2}{4} (M^2 \vec{S})^i D_x S^j S^k \right] .$$  (2.39)

Higher conserved charges $Q_r$, where the subindex $r$ indicates that it contains at most $r$ derivatives of the fields, are recursively generated from the knowledge of the Hamiltonian, as follows

$$Q_{r+1} = \{ B[H], Q_r \} .$$  (2.40)

This formula is the analogue of (2.32), where in the continuum limit the commutator has been replaced by the Poisson brackets. These higher charges Poisson commute among each others, and therefore are conserved, namely

$$0 = \frac{\mathrm{d}Q_r}{\mathrm{d}t} , \qquad \{ Q_p, Q_q \} = 0 , \qquad \forall\, p, q .$$  (2.41)

Furthermore, the charges $Q_2 \equiv H$ and $Q_3$ satisfy the following continuity equation,

$$\left. \left( \frac{\mathrm{d}\mathscr{Q}_2}{\mathrm{d}t} - \frac{\mathrm{d}\mathscr{Q}_3}{\mathrm{d}x} \right) \right|_{\text{on-shell}} = 0 ,$$  (2.42)

where

$$Q_r \equiv \int_0^L \mathrm{d}x \, \mathscr{Q}_r , \qquad \mathscr{Q}_2 \equiv \mathscr{H} .$$  (2.43)

Below we give a proof of formula (2.42), which holds as a direct consequence of the fact that the charge $Q_2$ is the Hamiltonian, namely the generator of time evolution.

*Proof.* From the definition of $Q_3$, we have that

$$Q_3 = \int \mathrm{d}x\mathrm{d}y\mathrm{d}z \left( x \frac{\delta \mathscr{Q}_2(x,t)}{\delta \varphi(z,t)} \frac{\delta \mathscr{Q}_2(y,t)}{\delta \pi(z,t)} - x \frac{\delta \mathscr{Q}_2(x,t)}{\delta \pi(z,t)} \frac{\delta \mathscr{Q}_2(y,t)}{\delta \varphi(z,t)} \right) ,$$  (2.44)

where we collectively denoted by $\varphi(x,t)$ and $\pi(x,t)$ the fields and their conjugate momenta, respectively. Summation over fields and momenta is left implicit. The functional derivatives will produce $\delta$-functions that will eventually identify $x, y, z$ to be the same variable. Therefore, when computing the spatial derivative of $\mathscr{Q}_3$, we need to include the

contribution from all these variables. This means the following

$$
\begin{aligned}
\frac{\mathrm{d}\mathscr{Q}_3}{\mathrm{d}x} \;=\; \int \mathrm{d}y\mathrm{d}z \bigg[ & \frac{\delta\mathscr{Q}_2(x,t)}{\delta\varphi(z,t)}\frac{\delta\mathscr{Q}_2(y,t)}{\delta\pi(z,t)} - \frac{\delta\mathscr{Q}_2(x,t)}{\delta\pi(z,t)}\frac{\delta\mathscr{Q}_2(y,t)}{\delta\varphi(z,t)} \\
& + x\frac{\mathrm{d}}{\mathrm{d}x}\left(\frac{\delta\mathscr{Q}_2(x,t)}{\delta\varphi(z,t)}\right)\frac{\delta\mathscr{Q}_2(y,t)}{\delta\pi(z,t)} - x\frac{\mathrm{d}}{\mathrm{d}x}\left(\frac{\delta\mathscr{Q}_2(x,t)}{\delta\pi(z,t)}\right)\frac{\delta\mathscr{Q}_2(y,t)}{\delta\varphi(z,t)} \\
& + x\frac{\delta\mathscr{Q}_2(x,t)}{\delta\varphi(z,t)}\frac{\mathrm{d}}{\mathrm{d}y}\left(\frac{\delta\mathscr{Q}_2(y,t)}{\delta\pi(z,t)}\right) - x\frac{\delta\mathscr{Q}_2(x,t)}{\delta\pi(z,t)}\frac{\mathrm{d}}{\mathrm{d}y}\left(\frac{\delta\mathscr{Q}_2(y,t)}{\delta\varphi(z,t)}\right) \bigg] \,.
\end{aligned}
\tag{2.45}
$$

The cross terms in the second and third line of the above expression cancel out between each others, if one reads it bearing in mind the identification of $x$ and $y$ coming from the $\delta$-functions. Then, by using the equations of motion:

$$
\frac{\mathrm{d}\varphi(x,t)}{\mathrm{d}t} = \frac{\delta H}{\delta\pi(x,t)}\,, \qquad\qquad \frac{\mathrm{d}\pi(x,t)}{\mathrm{d}t} = -\frac{\delta H}{\delta\varphi(x,t)}\,,
\tag{2.46}
$$

we can rewrite the first line of (2.45) as $\mathrm{d}\mathscr{Q}_2/\mathrm{d}t$, therefore proving (2.42). □

For the LL theory, this construction gives the infinite tower of conserved charges required by integrability.[6] For the Class 5 model (2.15) this construction still works, and we checked it up to (and including) $Q_4$. Actually, for the reasoning that we discuss in the next section, we could have stopped at checking the existence of $Q_3$, since this is enough to guarantee the existence of the whole tower of higher conserved charges.

## 2.3 Strong symmetry and the generation of higher conserved charges

In [55] it was shown that, for the regular LL action, $B[H]$ (called $\bar{T}(S)$ in the reference) generates the conserved densities of the theory. Although we are working with a deformed version of the LL action, we can apply the same argument step by step to show that $B[H]$ generates conserved quantities of the deformed theory.

The argument is divided into two steps: first, defining the conserved charges iteratively using the boost operator $Q_{r+1} = \{B[H], Q_r\}$ and assuming $\{Q_r, Q_{r+1}\} = 0$, we can use Jacobi identity of the Poisson brackets to iteratively show that $\{Q_r, Q_s\} = 0$ with $s > r$

$$
\begin{aligned}
\{Q_r, Q_{s+1}\} = \{Q_r, \{B[H], Q_s\}\} &= -\{B[H], \{Q_s, Q_r\}\} - \{Q_s, \{Q_r, B[H]\}\} \\
&= -\{B[H], \{Q_s, Q_r\}\} + \{Q_s, Q_{r+1}\}\,.
\end{aligned}
\tag{2.47}
$$

Thus, the second step is to close the induction argument by proving that $\{Q_r, Q_{r+1}\} = 0$.

---

[6]In the $(\theta, \phi)$ variables, it is immediate to check that the higher charges are conserved and Poisson commute among each others. In the spin density variables $\vec{S}$, this still holds true, but only after imposing the constraint $|\vec{S}|^2 = 1$.

This is proven by analysing which is the leading term of this expression. It is easy to see that, because $\mathscr{H}$ is quadratic in $S$ and involves two derivatives with respect to $x$, then the iterative definition of the conserved charges imposes that the leading order of $\mathscr{Q}_r$ involves $r$ powers of $S$ and $r$ derivatives with respect to $x$, and it is invariant under the $O(3)$ rotation of the spin variables. Similarly, we can argue that the leading order of $\{Q_r, Q_{r+1}\}$ has to involve $2r$ powers of $S$ and $2r + 1$ derivatives with respect to $x$. This term can only come from the Poisson bracket of the leading terms of $Q_r$ and $Q_{r+1}$. As the leading terms are the same in both our case and the undeformed one, we can directly use the proof of [55] to claim that $\{Q_r, Q_{r+1}\} = 0$.

This is enough to show that all the quantities obtained from $B[H]$ are conserved quantities in involution. Requiring the existence of an operator that is capable of creating a hierarchy of conserved quantities is an alternative method to requiring a Lax pair, and it is equally powerful. There is a large amount of literature dedicated to such operators, but nearly all of it is written in terms of symmetries, i.e. vector fields, instead of conserved quantities. Nevertheless, both languages are equivalent. If we define the vector field $X_A$ as

$$X_A(B) = \{A, B\} \,, \tag{2.48}$$

we have the following isomorphism between Lie commutators and Poisson brackets

$$X_{\{A,B\}} = [X_A, X_B]_L \,. \tag{2.49}$$

We want to finish this section by collecting some concepts regarding recursion operator. Here we compile results from [56–60].

Given a differential equation

$$u_t + K(x, u, u_x, u_{xx}, \dots) = 0 \,, \tag{2.50}$$

where $K(x, u, u_x, u_{xx}, \dots)$ is a function of the position $x$, the field $u$ and its spatial derivatives. For brevity, we will write just $K(u)$ from now on. Then, we say that the vector $\xi$ is a *symmetry* of this differential equation if

$$\frac{\mathrm{d}\xi}{\mathrm{d}t} + K'(u)[\xi] = \frac{\partial \xi}{\partial t} - \xi'[K(u)] + K'(u)[\xi] = 0 \,, \tag{2.51}$$

where the prime denotes the Fréchet derivative, defined as

$$K'(u)[\xi] \equiv \lim_{\epsilon \to 0} \frac{\partial}{\partial \epsilon} K(u + \epsilon \xi) \,. \tag{2.52}$$

The Fréchet derivatives can be written in terms of the associated vector fields as $K'(u)[\xi] =$

$X_\xi K(u)$, so the above condition can be rewritten as

$$\frac{\partial \xi}{\partial t} + X_\xi K(u) - X_{K(u)}\xi = 0\,. \tag{2.53}$$

If $\xi$ is independent of time and $K$ can be written as a vector field, $K = X_{\mathcal{H}}$, then $\xi$ is a symmetry if the two flows commute $[X_\xi, X_{\mathcal{H}}]_L = 0$.

We say that $\mathscr{R}$ is a *strong symmetry*, or a *recursion operator*, if it maps symmetries into symmetries. It can be checked that an operator $\mathscr{R}$ that fulfils

$$(\mathscr{R}(\xi))'[K(u)] - K'(u)[\mathscr{R}(\xi)] = \mathscr{R}(\xi'[K(u)]) - \mathscr{R}(K'(u)[\xi])\,. \tag{2.54}$$

is in particular a strong symmetry.[7] A weaker definition of a recursion operator can be found, for example, in [59]

$$\frac{\mathrm{d}\mathscr{R}(\xi)}{\mathrm{d}t} + K'(u)[\mathscr{R}(\xi)] = \hat{\mathscr{R}}\left(\frac{\mathrm{d}\xi}{\mathrm{d}t} + K'(u)[\xi]\right)\,, \tag{2.55}$$

where $\mathscr{R}$ and $\hat{\mathscr{R}}$ are both linear operators. From this definition, it is obvious that if $\xi$ is a symmetry of the equations of motion, then $\mathscr{R}(\xi)$ is also a symmetry.

We say that $\mathscr{R}$ is a *hereditary operator*, or *Nijenhuis operator*, if $[\mathscr{R}', \mathscr{R}]$ is symmetric, that is, if

$$[\mathscr{R}', \mathscr{R}][v](w) = \mathscr{R}'[v](\mathscr{R}(w)) - \mathscr{R}(\mathscr{R}'[v](w)) = \mathscr{R}'[w](\mathscr{R}(v)) - \mathscr{R}(\mathscr{R}'[w](v)) = [\mathscr{R}', \mathscr{R}][w](u)\,. \tag{2.56}$$

If $\mathscr{R}$ is a linear operator, the condition of hereditary operator takes the following simpler form

$$\mathscr{R}^2([\xi, \rho]_L) + [\mathscr{R}(\xi), \mathscr{R}(\rho)]_L = \mathscr{R}([\xi, \mathscr{R}(\rho)]_L) + \mathscr{R}([\mathscr{R}(\xi), \rho]_L)\,. \tag{2.57}$$

The main theorem behind hereditary operators is the fact that, if $\mathscr{R}$ is a hereditary operator and a strong symmetry with respect to the differential equation $u_t + K = 0$, then it is also a strong symmetry with respect to the differential equation $u_t + \mathscr{R}(K) = 0$.

In the language of conserved charges, a strong symmetry takes a charge $Q$ that commutes with the Hamiltonian $H$ and gives us another charge $\tilde{Q}$ that also commutes with $H$. However, this condition does not guarantee that $\{Q, \tilde{Q}\} = 0$. Only if $\mathscr{R}$ is both a strong symmetry and a hereditary operator, it is guaranteed that $\{Q, \tilde{Q}\} = 0$.

It is also interesting to comment that a strong symmetry can be used to write a system

---

[7]The converse is only true if the differential equation (2.50) is regular [56]. A differential equation is called regular if it has a unique solution $u(t, u_0)$ for every time $t$ and every initial condition $u(t_0) = u_0$, and this solution is differentiable with respect to $u_0$.

of equations similar to a Zakharov-Shabat formulation

$$\begin{cases} \mathscr{R}\psi = \lambda\psi \\ \frac{d\psi}{dt} + K'(u)[\psi] = 0 \end{cases} \tag{2.58}$$

# 3  Particle production?

To perform a field theory analysis, it is more useful to rewrite the action (2.13) in terms of a single complex scalar field instead of the angles $\theta$ and $\phi$. If we define the field [61]

$$\eta = \frac{\sin\theta}{\sqrt{2 + 2\cos\theta}} e^{i\phi}, \tag{3.1}$$

and add a total derivative term $\epsilon^2\phi_t$, the kinetic term of the Lagrangian takes the standard form

$$\mathcal{L}_5^\eta = i(\eta_t^*\eta - \eta^*\eta_t) - V(\eta^*, \eta, \eta_x^*, \eta_x), \tag{3.2}$$

where $V$ is a potential that has a very involved that is not very illuminating. The field redefinition might not seem worth, but the quantisation and perturbation theory of this action becomes much simpler, see for example [62, 63]. If we expand this action in the number of fields, we get[8]

$$\mathcal{L}_5^\eta = -\alpha\eta_x^* + \left( i(\eta_t^*\eta - \eta^*\eta_t) - 2\eta_x^*\eta_x \right) + \frac{1}{2}\alpha\eta^*\left( 2\eta_x^*\eta - 3\eta^*\eta_x \right) \\ - \left( (\eta_x^*)^2\,\eta^2 + (\eta^*)^2\,(\eta_x)^2 \right) + \mathcal{O}(\eta^5). \tag{3.3}$$

We can see that, in contrast with the usual LL action, this one has cubic terms in the fields, which has major consequences.[9] The linear term is irrelevant, as it is a total derivative.

As the quadratic part of the Lagrangian is the same as in the undeformed LL, the discussion of [62] regarding propagator and dispersion relation is unaffected. The relevant pieces of that discussion are that the action is linear in time derivatives, so the field operator in the interaction picture only contains negative-frequency modes

$$\eta(t, x) = \int \frac{dp}{2\pi} b_p e^{ipx - i\omega t}, \qquad \eta^*(t, x) = \int \frac{dp}{2\pi} b_p^\dagger e^{-ipx + i\omega t}, \tag{3.4}$$

with $b_p$ and $b_p^\dagger$ annihilation and creation operators respectively. In addition, the ground

---

[8]Instead of expanding the action (2.13), we may have expanded the equivalent action (2.20) which has no derivatives in the interaction term. However, that action expressed in the $\eta$ variable (3.1) produces in the field expansion a linear term that is not a total derivative, meaning the expansion is not about a solution of the equations of motion. A similar issue is described in Appendix A for the Class 6 model.

[9]Using the method described in appendix A of [25], it can be checked that the cubic term cannot be eliminated through a field redefinition.

state is annihilated by $\eta(t, x)$ and the propagator only has one pole in momentum representation. From the equation of motion for $\eta^*$ is immediate to get the dispersion relation $\omega = -p^2$.[10]

We are now in a position to compute the S-matrix of this Lagrangian in perturbation theory, defined as

$$\mathcal{S}(\text{in}; \text{out}) = \langle \text{out}|T e^{-i \int dt\, dx\, V}|\text{in}\rangle. \tag{3.5}$$

As our potential is independent of $x$ and $t$, the integral over these two variables gives rise to conservation of energy and momentum. We will focus here on the case of one incoming and two outgoing particles, where the delta function that enforce this conservation can be rewritten as

$$\delta(\omega(k) - \omega(p) - \omega(p'))\delta(k - p - p') = \frac{\delta(p)\delta(k - p')}{|2p'|} + \frac{\delta(p')\delta(k - p)}{|2p|}. \tag{3.6}$$

Thus, the tree-level contribution to the one-to-two S-matrix is given by

$$
\begin{aligned}
\mathcal{S}(k; p, p') &= -i \int dt\, dx\, \langle 0|b_p b_{p'} \frac{2\alpha \eta^*(t, x)\eta_x^*(t, x)\eta(t, x) - 3\alpha[\eta^*(t, x)]^2 \eta_x(t, x)}{2} b_k^\dagger|0\rangle \\
&= -i \left(\alpha(p + p') - \frac{3}{2}\alpha k\right)\left(\frac{\delta(p)\delta(k - p')}{|2p'|} + \frac{\delta(p')\delta(k - p)}{|2p|}\right) \\
&= \frac{i\alpha k}{4|k|}\left[\delta(p)\delta(k - p') + \delta(p')\delta(k - p)\right].
\end{aligned}
\tag{3.7}
$$

From that, we can see that one excitation of the field $\eta$ with arbitrary momentum can decay into two excitations of the same field, as long as one of them has zero momentum. Although we may think that having particle production is prohibited by integrability, this requires massive particles [10][11] and breaks down in the case of massless 2d scalar scattering, see [12, 13] and references therein.

Regarding the tree-level $2 \to 2$ S-matrix, because the four-field term is not deformed, and the Lagrangian does not contain the $\eta \times \eta \to \eta$ vertex, it has the same form as in the undeformed case.

This problem can be attacked from the same perspective as QED. The problem with computing matrix elements in the interaction picture, both in QED and here, is the existence of terms in the interaction potential that do not decay at early time or late time, $|t| \to \pm\infty$, and therefore create IR divergences. The solution proposed by Faddeev and Kulish (FK) [64] was to move those terms from the interaction potential into the states, so the soft particles appear inside the asymptotic states instead of on the S-matrix.[12] As

---

[10]The proportionality constant between $\omega$ and $p^2$ is unphysical, as we can change it by performing the rescaling $t \to \alpha t$ and $x \to \alpha x$ for any value of $\alpha$.

[11]This proof explicitly uses the assumption that wave packets can be separated, which is not possible for massless particles.

[12]Technically speaking, this is not a standard S-matrix, as the FK asymptotic states do not belong to

the cubic term in (3.3) appears only due to the deformation, and therefore it is the only term that does not die off at late time, it is immediate to see that the transformation of the states needed to eliminate it from the interaction potential is the continuum limit of the Drinfeld twist (1.9).[13]

Although surprising, the features of this model are not that strange in the context of integrable models. First and foremost, tree-level particle production is found in several classically integrable models, like the [65–67] or a generalised $O(N)$ $\sigma$-model [68]. Second, cubic vertices are ubiquitous in integrable theories, but they usually do not create any problems either because the particles are massive (and thus their decay is forbidden) or because the particle appears as a bound state of itself in the scattering process (as happens in the Bullough-Dodd model [69]). Finally, the fact that the Lagrangian is not Hermitian, so it encodes the process $\eta \to \eta \times \eta$, but it does not encode the inverse process. A well-known example of a theory with similar characteristic is the double scaling limit of $\gamma$-deformed planar $\mathcal{N} = 4$ SYM, usually called fishnet theory [70].

The situation we describe here is very reminiscent of the Jordanian Yang-Baxter deformation studied in [47], where the authors of that article also start from a deformation of an integrable model that do not destroy its integrability and, after considering a perturbative QFT, they find the presence of cubic vertices that lead to particle production. However, in contrast to our model, the excitations they create are either massive or massless but non-soft. Nevertheless, as Yang-Baxter deformations on the string side are related to Drinfeld twists on the field theory side in the context of $\mathrm{AdS}_5/\mathrm{CFT}_4$, both the system described in [47] (see also [46]) and the one described here have striking similarity that may lead to an explanation: the Hamiltonian associated to the spin chain is non-diagonalisable, with states with different number of excitations leaving in the same Jordan chain.

In our situation, we can refine more this statement. The global $\mathfrak{su}(2)$ symmetry of the Heisenberg XXX spin chain is broken when we turn on the $a$ deformation of the Class 5 model, leaving $\sum \mathfrak{s}_n^+$ as the only unbroken generator. Thus, the fact that there is no conserved quantum number associated to the number of excitations (which, in our case, can be identified with $\sum (\frac{1}{2} + \mathfrak{s}_n^z)$) is a clear hint that the integrable field theory associated to the continuum limit of such model might have particle production. In addition, the fact that $\sum \mathfrak{s}_n^+$ is still a symmetry allows us to write a one-to-one correspondence between Jordan chains of generalised eigenstates of the Class 5 model and descendants states of the Heisenberg XXX spin chain (see [71] for a more in depth explanation). As a consequence, the time evolution is capable of creating particles with zero momentum, similarly to how $\sum \mathfrak{s}_n^+$ creates descendant states. It is worth comparing this with the quantisation of the Kadomtsev-Petviashvili equation [11], where the canonical quantisation leads to an

---

a Fock space, but to a "coherent state space".

[13]This identification between the FK states and states of the undeformed LL theory is only formal, since the FK states are dressed by a cloud of soft particles, whereas the others are not.

integrable Hamiltonian that does not preserve the number of particles but preserves the mass. Thus, the model presents particle creation, but it only can relate states with the same total mass.

# 4    Conclusions

In this article, we constructed the 2d sigma models associated with the two possible non-diagonalisable integrable deformations of the XXX spin chain that appeared in the classification of [1], namely the "Class 5" and "Class 6" models. We analysed various properties of these models, such as their higher conserved charges, their spacetime symmetries and whether they admit particle production. There are still a number of questions that would be interesting exploring for the future.

**Lax pair.**    Showing that the equations of motion admit a Lax pair is a typical route to construct the higher conserved charges eventually needed to show integrability. In this work, we directly constructed the higher conserved charges via the boost method, bypassing the need to find a Lax pair. However, since the Lax pair for the LL model is known [72], see also e.g. [73], we expect that the models considered in this article also admit a Lax pair. Although not straightforward to guess, we expect their Lax pairs to be a deformation of the one already constructed for the LL action.

**Drinfeld twist.**    In this article, we showed that the Class 5 R-matrix is realised as a Drinfeld twist of the XXX spin chain. This suggests that this model might play a role in the deformed version of the AdS/CFT, where a Drinfeld twist of the XXX spin chain appearing in the $\mathcal{N} = 4$ super Yang-Mills corresponds to a Jordanian deformation of the AdS$_5 \times$S$^5$ superstring background, see [46] for a recent work. Then, in light of this, it would be interesting to see if there is any limit of the Jordanian deformed superstring action that reduces to the action (2.13), similarly to what happened for the undeformed AdS$_5 \times$S$^5$ superstring [24, 25, 74].

**Non-relativistic limits.**    Another instance where the LL action appears in string theory is via non-relativistic limits. It has been shown that by taking a non-relativistic limit both on target space and world-sheet of the AdS$_5 \times$S$^5$ superstring action produces a LL model [75].[14] It would be interesting to explore whether the integrable deformations of the LL actions constructed in this article emerge as a non-relativistic limit of a suitably deformed AdS$_5 \times$S$^5$ superstring action.

---

[14]This result has been generalised to groups others than $SU(2)$, see e.g. [76, 77].

**Classifying all integrable deformations of the LL model.** It would be interesting to use the boost functional to put some constrains on a generic deformation ansatz of the LL action, i.e. construct a $Q_3$ and demand to commute with the Hamiltonian, leading to a classification of all possible integrable deformations. It would be interesting to see if this procedure gives exactly the Class 5 and Class 6 models of [1], or if there are more possibilities.[15]

**Solitons.** Integrable models typically admit a soliton solution, although this is not a necessary, neither sufficient, condition to prove integrability. It is well-known that the LL theory admits a soliton solution [79–81]. It would be interesting to explore if the soliton survives in the presence of the integrable deformations considered in this paper. From few attempts, it seems very important to identify the correct boundary conditions, as otherwise the soliton solution breaks down.

**Quantum analogue of strong symmetries.** The fact that the boost functional acts as a generator of strong and hereditary symmetry has the implication that the existence of $Q_3$ is enough to guarantee the existence of the infinite tower of higher conserved charges. This result holds thanks to Fuchssteiner's theorem [56], and it is valid at the level of the classical field theory. It would be important to understand if there is an analogue of this statement that holds at the level of the R-matrix describing the quantum spin chain.

**Particle production.** One may wonder whether the particle production allowed by the Class 5 model is physical, since the additional particle appearing in the $1 \to 2$ process has zero momentum. This problem was already considered by Weinberg for the collision of gravitons [82], where he showed that any soft graviton that appears in asymptotic states can at most contribute to the S-matrix with a phase factor. It was discovered later that this seemingly innocuous phase factor is actually physical, as it is equivalent to the appearance of BMS supertranslations as asymptotic symmetries in flat spacetime [83]. We wonder if the soft particle production found in this article for the Class 5 model is of any physical relevance.

**S-matrix factorization.** Factorization of the $3 \to 3$ scattering is not guaranteed for theories which are not Lorentz invariant and where the particles involved are massless. Despite that, the LL theory still shows factorisability of the S-matrix, as it can be found from an explicit computation [84]. It would be interesting to check if factorisability holds for the Class 5 model.

---

[15]A similar procedure, although with a different method, has been performed in [78]. Because of a complicated change of variables, these results are not immediately accessible.

# Acknowledgments

We would like to thank G. Arutyunov, R. Borsato, L. Corcoran, S. Driezen, S. Frolov, B. Hoare, A. Holguin, R. Ruiz and A. Torrielli for interesting discussions. We would like to thank C. Paletta for pointing us the article [55]. We also thank S. Driezen, R. Hernandez, R. Ruiz and A. Torrielli for useful comments on a draft of this manuscript. MdL was supported in part by SFI and the Royal Society for funding under grants UF160578, RGF\ R1\ 181011, RGF\8EA\180167 and RF\ ERE\ 210373. MdL is also supported by ERC-2022-CoG - FAIM 101088193. AF is supported by the SFI and the Royal Society under the grant number RFF\EREF\210373. JMNG is supported by the Deutsche Forschungsgemeinschaft (DFG, German Research Foundation) under Germany's Excellence Strategy – EXC 2121 "Quantum Universe" – 390833306 and by the Deutsche Forschungsgemeinschaft (DFG, German Research Foundation) – SFB-Geschäftszeichen 1624 – Projektnummer 506632645. AF thanks Lia for her permanent support.

# Appendices

## A  The Class 6 model of [1]

The Class 6 model is the second model that appeared in the classification of [1] that describes a continuum deformation of the XXX spin chain. It is expressed in terms of the R-matrix[16]

$$R_6(u) = \begin{pmatrix} 1 + 2a_1u & a_2u(1 + 2a_1u) & a_2u(1 + 2a_1u) & -a_2^2u^2(1 + 2a_1u)^2 \\ 0 & 2a_1u & 1 & -a_2u(1 + 2a_1u) \\ 0 & 1 & 2a_1u & -a_2u(1 + 2a_1u) \\ 0 & 0 & 0 & 1 + 2a_1u \end{pmatrix}, \quad (A.1)$$

which is non-diagonalisable, and it satisfies the Yang-Baxter equation (1.4). The R-matrix depends on two free parameters $a_1, a_2$; however, similarly to the Class 5 model, the parameter $a_1$ can be eliminated by the redefinition $u \to u/a_1$, $a_2 \to a_1a_2$. Therefore, from now on, we set $a_1 = 1$ and we denote the single deformation parameter by $a \equiv a_2$.

The quantum Hamiltonian associated to $R_6$ is the following:

$$\mathcal{H}_6 = \sum_{j=1}^{L} h_{j,j+1} , \qquad h_{L,L+1} = h_{L,1} , \qquad (A.2)$$

$$h_{j,j+1} = 2P_{j,j+1} + \frac{a}{2}\left(\sigma_j^z \otimes \sigma_{j+1}^+ + \sigma_j^+ \otimes \sigma_{j+1}^z\right) = \begin{pmatrix} 2 & a & a & 0 \\ 0 & 0 & 2 & -a \\ 0 & 2 & 0 & -a \\ 0 & 0 & 0 & 2 \end{pmatrix}_{j,j+1} ,$$

In this case the $R$-matrix again satisfies a Drinfeld twist like factorization

$$R_6(u) = F_{12}(u)R_{12}^{XXX}(u)F_{21}^{-1}(-u), \quad F = \begin{pmatrix} 1 & \frac{a}{2}(u-1)u & \frac{a}{2}(u-1)u & 0 \\ 0 & 1 & 0 & \frac{a}{2}(1-u)u \\ 0 & 0 & 1 & \frac{a}{2}(1-u)u \\ 0 & 0 & 0 & 1 \end{pmatrix}$$

$$(A.3)$$

However, in this case $F$ is a function of the rapidities and we were unable to express it in terms of symmetry generators in order to check the cocycle condition (1.10).

---

[16]The R-matrix given in (A.1) is the Class 6 R-matrix of [1] multiplied by $(1 - a_1u)^{-1}$, which amounts to shifting the Hamiltonian by a multiple of the identity operator, and is therefore unphysical.

## A.1 The continuum limit

The continuum limit for the Class 6 model follows the same steps that we took for the Class 5 model in section 2. The sigma model action is described by the analogue of formula (2.4), where $h_{12}$ is now given in (A.2), and gives the following result

$$S_6 = \int \mathrm{d}t \mathrm{d}x \left[ 1 + a \left( \mathcal{A}^{(0)} + \varepsilon \mathcal{A}^{(1)} + \varepsilon^2 \mathcal{A}^{(2)} \right) + \varepsilon^2 \mathcal{L}_{\mathrm{LL}} + \mathcal{O}(\varepsilon^3) \right]. \tag{A.4}$$

where

$$\mathcal{A}^{(0)} = -\frac{e^{-i\phi}}{2} \sin 2\theta, \tag{A.5}$$

$$\mathcal{A}^{(1)} = -\frac{e^{-i\phi}}{2} \left( \cos 2\theta \, \theta_x + \frac{i}{2} \sin 2\theta \, \phi_x \right), \tag{A.6}$$

$$\mathcal{A}^{(2)} = \frac{e^{-i\phi}}{4} \left( \sin 2\theta \, (\theta_x)^2 + 2i \cos^2 \theta \, \theta_x \phi_x + \frac{1}{2} \sin 2\theta \, (\phi_x)^2 \right.$$
$$\left. - \cos 2\theta \, \theta_{xx} + \frac{i}{2} \sin 2\theta \, \phi_{xx} \right), \tag{A.7}$$

$$\mathcal{L}_{\mathrm{LL}} = -\cos \theta \, \phi_t - \frac{1}{2} \left( (\theta_x)^2 + \sin^2 \theta (\phi_x)^2 \right). \tag{A.8}$$

The term $\mathcal{A}^{(1)}$ turns out to be a total derivative, and therefore, together with the constant term, can be disregarded. At this stage, we want to bring (A.4) into the form of a homogeneous action of order $\varepsilon^2$. To do that, we rescale the deformation parameter as $a \equiv \varepsilon^2 \alpha$. By doing this, we find that the continuum limit of the Class 6 R-matrix of [1] is the leading term of the action (A.4) in the $\varepsilon \to 0$ limit:

$$S_6^{\mathrm{lead}} = -\varepsilon^2 \int \mathrm{d}t \mathrm{d}x \left[ \cos \theta \, \phi_t + \frac{1}{2} \left( (\theta_x)^2 + \sin^2 \theta (\phi_x)^2 \right) + \frac{\alpha}{2} e^{-i\phi} \sin 2\theta \right]. \tag{A.9}$$

This action describes a LL theory in a non-unitary potential. Its off-shell spacetime symmetries are the same as the ones we found for the continuum limit of the Class 5 model, namely just time and space translations: $H, P$. The anisotropic dilatation (2.29), which is an on-shell symmetry of the LL theory, is again broken by the presence of the non-unitary potential.

Similarly to the Class 5 model, one can introduce the spin density variables given below equation (2.13), such that the classical Hamiltonian associated to the action (A.9) becomes

$$H_6 = \int \mathrm{d}x \left[ \frac{1}{2} \vec{S}_x^T \vec{S}_x + \frac{\alpha}{2} \vec{S}^T N \vec{S} \right], \qquad N \equiv \begin{pmatrix} 0 & \beta_1 & 1 + \beta_2 \\ -\beta_1 & 0 & -i + \beta_3 \\ 1 - \beta_2 & -i - \beta_3 & 0 \end{pmatrix}, \tag{A.10}$$

where $\beta_1, \beta_2, \beta_3$ are free parameters, which are spurious because they parametrize the antisymmetric part of $N$, and therefore give zero contribution in $\vec{S}^T N \vec{S}$. From now on, we set them to zero. Using the Poisson brackets (2.21), and the above classical Hamiltonian, the equations of motion are

$$\vec{S}_t = \vec{S} \times \left( \vec{S}_{xx} - \alpha N \vec{S} \right) . \tag{A.11}$$

It is interesting to note that, similarly to the case of the Class 5 model, the mass matrix is non-diagonalisable and has Jordan normal form with 0s everywhere and 1 on two entries in the upper diagonal

$$N = \begin{pmatrix} 0 & 0 & 1 \\ 0 & 0 & -i \\ 1 & -i & 0 \end{pmatrix} = \begin{pmatrix} i & 0 & i \\ 1 & 0 & 0 \\ 0 & i & 0 \end{pmatrix} \begin{pmatrix} 0 & 1 & 0 \\ 0 & 0 & 1 \\ 0 & 0 & 0 \end{pmatrix} \begin{pmatrix} 0 & 1 & 0 \\ 0 & 0 & -i \\ -i & -1 & 0 \end{pmatrix} . \tag{A.12}$$

For this reason, this model cannot be related by a rotation to the anisotropic LL theory.

### A.1.1   Higher conserved charges

We can follow the same logic that we presented in section 2.2 to construct the higher conserved charges for the Class 6 model in the continuum limit. The Hamiltonian in $(\theta, \phi)$ variables is given by

$$H = \int_0^L \mathrm{d}x \, \mathscr{H} , \tag{A.13}$$

$$\mathscr{H} = \frac{1}{2} \left[ (\theta_x)^2 + \sin^2 \theta (\phi_x)^2 \right] + \frac{\alpha}{2} e^{-i\phi} \sin 2\theta , \tag{A.14}$$

and the boost is defined again by the equation (2.35). The higher charges are generated with the Poisson commutation of a lower conserved charge, like the Hamiltonian, with the boost generator, i.e. formula (2.40). The charges that we generate with this procedure are again Poisson commuting among each others, and therefore conserved.

As an example, we give the expression for $Q_3$:

$$Q_3 = \{B[H], H\} = \int_0^L \mathrm{d}x \, \mathscr{Q}_3 , \tag{A.15}$$

$$\mathscr{Q}_3 = \frac{1}{2} \left[ 4\cos\theta \, (\theta_x^2 \phi_x) + \sin\theta \left( \sin(2\theta) \, \phi_x^2 - 2\,\theta_{xx} \right) \phi_x + 2\sin\theta \, (\theta_x \phi_{xx}) \right]$$
$$+ i\alpha e^{-i\phi} \left( \cos\theta \, \theta_x - i\cos(2\theta) \, \sin\theta \, \phi_x \right) . \tag{A.16}$$

and the analogue in spin variables is:

$$\mathcal{Q}_3 = \varepsilon_{ijk} \left[ S_x^i S_{xx}^j S^k + \alpha \, S_x^i (N\vec{S})^j S^k \right] . \tag{A.17}$$

### A.1.2 The $\eta$ field expansion

If we perform the same field redefinition as in (3.1), and we expand the Class 6 action (A.9) in powers of $\eta$, we get

$$\mathcal{L} = 2\alpha \, \eta^* + \left[ 2\, \eta_x^* \, \eta_x + i \left( -\eta \, \eta_t^* + \eta^* \, \eta_t \right) \right] + \left[ -5\alpha \, (\eta^*)^2 \, \eta \right] + \left[ \eta^2 \, (\eta_x^*)^2 + (\eta^*)^2 \, (\eta_x)^2 \right] + \ldots \tag{A.18}$$

The linear term appears because $\theta = $ const. and $\phi = $ const. is not a solution of the equations of motion. This means the correct vacuum to consider is a function of the spacetime coordinates, which will in turn make the expanded Lagrangian depending on the spacetime coordinates as well. The treatment of this type of Lagrangians is typically quite involved, and therefore we do not investigate the particle production any further.

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
