# Peer review of "An integrable deformed Landau-Lifshitz model with particle production?"

_SciPost Physics_

## Round 1 · Referee Report · Anonymous (Referee 1) · 2025-10-4

Report

I recommend that this paper be accepted for publication in this journal.

The authors have written a very interesting paper on an integrable, but non-unitary deformation of the XXX model. They then consider the continuum limit of this theory to find its classical analog. While not quite proving that the classical theory is integrable, they are able to construct an infinite tower of conserved quantities in involution, highly suggesting that the theory is integrable and a Lax pair can be constructed. Perhaps most interesting, they show that the theory can have 1 --> 2 particle production, where one of the two outgoing particles has zero momentum, which is a consequence of momentum conservation.

The paper is clearly written, and as I best as I can tell, free of any errors. I expect there to be a wide interest in the paper. For these reasons I recommend that it be published.

Recommendation

Publish (easily meets expectations and criteria for this Journal; among top 50%)

---

## Round 1 · Referee Report · Anonymous (Referee 2) · 2025-11-17

Disclosure of Generative AI use

The referee discloses that the following generative AI tools have been used in the preparation of this report:

This report was prepared with the assistance of ChatGPT 5.1. The tool was used exclusively to refine grammar, improve clarity, and standardise the formulation of sentences. All scientific judgements, critical assessments, and final editorial decisions expressed in the report are entirely ours. No confidential information about the manuscript was shared with the tool beyond the text reproduced in this report.

Report

We do not believe that the manuscript, in its current state, opens a new pathway in an existing or a new research direction with clear potential for substantial follow-up work. Unless the authors satisfactorily address the critiques raised in the attached referee report, we cannot recommend the publication of the manuscript.

Requested changes

-

Attachment

Recommendation

Ask for major revision

  • validity: -
  • significance: -
  • originality: -
  • clarity: -
  • formatting: -
  • grammar: -

Author:  Juan Miguel Nieto García  on 2025-12-18  [id 6157]

(in reply to Report 2 on 2025-11-17)
Category:
answer to question
reply to objection

Our answer to the referee's comments are in the attached file.

Attachment:

Comments.pdf

---

## Round 1 · Referee Report · Anonymous (Referee 3) · 2025-11-24

Report

The present article investigates the continuum coherent–state limit of the non-Hermitian integrable deformation of the Heisenberg XXX spin chain that appeared in the classification of $ 4 \times 4 $ Yang–Baxter solutions, referred to as Class 5 and 6. One of the key results is that the Class 5 $R$-matrix is not a genuinely new integrable object, but can be given by a Jordanian Drinfeld twist of the XXX $R$-matrix. In the continuum limit with appropriate scaling of the deformation parameter, the authors derive a leading classical field-theoretic action – non-unitary deformation of the Landau–Lifshitz model (LL). In spin-density basis this yields a Hamiltonian containing a non-diagonalisable “mass/gauge” structure that cannot be rotated to standard anisotropic LL model. Although an explicit Lax pair was not provided, the classical integrability is demonstrated by producing an infinite hierarchy of local conserved charges in involution, generated recursively by continuum analogue of the boost operator. The integrability criterion is based on the boost action as a strong symmetry/recursion operator. Authors point out the emerging symmetries and provide an argument for the on-shell symmetry breaking, i.e. anisotropic dilatation is absent due to the deformation.

By analysing LL scattering in the complex field formulation they find deformation introduces cubic vertices that were absent in the undeformed LL model. A tree-level computation then revealed a nonzero $1 \rightarrow 2$ amplitude, implying a particle decay into two (with soft excitation), which appears to be consistent with FK asymptotic dressing and its relation to Drinfeld twist. Hence authors have established a new example of an integrable deformed model that allows for particle “production”, which is tied to the spin chain non-diagonalisability and $\mathfrak{su}(2)$ breaking down to a single generator, yet preserving integrability. It is indeed an important step towards better understanding of the non-Hermitean integrable models, since it is usually neglected due to its unclear physical properties. In fact, this sector of integrable models and structures emerging therein arise in a number of subfields. For example, it would be particularly important to investigate relation to higher rank LL models, their deformations and lattice realisations, which also admit limits to various dynamical integrable systems (e.g. integrable subsector of Landau-Lifshitz-Gilbert models). Moreover such lower dimensional non-Hermitean structures play a key role in identifying new Poisson algebraic structures on unipotent groups that characterise Hamilton flows in higher quantum simplices. As it was also noted, another crucial question would be to identify a quantisation scheme for strong symmetries and explore how recursive structure translates to the lattice level. The last would be also useful for proving saturation conditions for involutive hierarchies from the quantum minimal surface perspective.

The article is self-consistent and presents clear results, backed by explicit propositions and derivations. I recommend it for publication. Some minor insignificant errata:

  1. “many of the unconventional characteristic”, p. 3, paragraph 1
  2. “furthers simplifications”, p. 4, paragraph below (1.1)
  3. “Galiean boost”, p. 11, footnote 4
  4. “once can check”, p. 12, footnote 5
  5. “commute among each others”, p. 12, below (2.32)
  6. “[R', R]w”, in the last equality of (2.56)
  7. “integrable model that do not destroy its integrability”, p. 19, paragraph 3
  8. “excitations leaving in the same Jordan chain”, p. 19, paragraph 3
  9. “descendants states”, p. 19, paragraph 4
  10. “groups others than SU(2)”, p. 20, footnote 14
  11. “some constrains”, p. 21, paragraph 1

Recommendation

Publish (easily meets expectations and criteria for this Journal; among top 50%)

  • validity: high
  • significance: high
  • originality: good
  • clarity: high
  • formatting: -
  • grammar: -

Author:  Juan Miguel Nieto García  on 2025-12-18  [id 6156]

(in reply to Report 3 on 2025-11-24)

Thank you for the comments, we fixed the typos in the new version.

---

## Round 2 · Author Response

We address the comments from referees 2 and fixed the typos pointed by referee 3.

---

## Round 2 · List of Changes

1. We have fixed the typos pointed by referee 3.
  2. We implemented the minor points raised by referee 2.
  3. We added footnote 6 above eq. (2.36) regarding the matching of classical and quantum charges.
  4. We added footnote 12 regarding the aim of computing the S-matrix.
  5. We have rewritten parts of section 2.3 to make it more clear.
  6. We amended the "S-matrix factorization" point in the conclusions.
  7. We have replaced $L$ by $\ell$ for the continuum model.

---

## Editorial Decision

refereeing_in_preparation